# Comparative NMR Metabolomics Profiling between Mexican Ancestral & Artisanal Mezcals and Industrialized Wines to Discriminate Geographical Origins, Agave Species or Grape Varieties and Manufacturing Processes as a Function of Their Quality Attributes

**DOI:** 10.3390/foods10010157

**Published:** 2021-01-13

**Authors:** Rosa López-Aguilar, Holber Zuleta-Prada, Arturo Hernández-Montes, José Enrique Herbert-Pucheta

**Affiliations:** 1Departamento de Ingeniería Agroindustrial, Universidad Autónoma Chapingo, km. 38.5 Carretera México-Texcoco, 56230 Chapingo, Estado de México, Mexico; rosloagg@gmail.com; 2Laboratorio de Productos Naturales, Área de Química, Departamento de Preparatoria Agrícola, Universidad Autónoma Chapingo, km. 38.5 Carretera México-Texcoco, 56230 Chapingo, Estado de México, Mexico; hozuleta_13@comunidad.unam.mx; 3Consejo Nacional de Ciencia y Tecnología-Laboratorio Nacional de Investigación y Servicio Agroalimentario Forestal, Universidad Autónoma Chapingo, 56230 Chapingo, Estado de México, Mexico; 4Departamento de Química Orgánica, Escuela Nacional de Ciencias Biológicas, Instituto Politécnico Nacional, Prolongación de Carpio y Plan de Ayala s/n, Colonia Santo Tomás, 11340 Ciudad de México, Estado de México, Mexico

**Keywords:** ancestral and artisanal mezcals, industrialized wines, proton nuclear magnetic resonance, supervised orthogonal projections to latent structures discriminant analysis

## Abstract

The oenological industry has benefited from the use of Nuclear Magnetic Resonance (^1^H-NMR) spectroscopy in combination with Multivariate Statistical Analysis (MSA) as a foodomics tool for retrieving discriminant features related to geographical origins, grape varieties, and further quality controls. Said omics methods have gained such attention that Intergovernmental Organizations and Control Agencies are currently recommending their massive use amongst countries as quality compliances for tracking standard and degradation parameters, fermentation products, polyphenols, amino acids, geographical origins, appellations d’origine contrôlée and type of monovarietal strains in wines. This study presents, for the first time, a ^1^H-NMR/MSA profiling of industrial Mexican wines, finding excellent statistical features to discriminate between oenological regions and grape varieties with supervised Orthogonal Projections to Latent Structures Discriminant Analysis (OPLS-DA). In a comparative way, it is applied with the ^1^H-NMR/OPLS-DA workflow for the first time in ancestral and artisanal Mexican mezcals with promising results to discriminate between regions, agave species and manufacturing processes. The central aim of this comparative study is to extrapolate the know-how of wine-omics into the non-professionalized mezcal industry for establishing the NMR acquisition, preprocessing and statistical analysis basis to implement novel, non-invasive and highly reproducible regional, agave species and manufacturing-quality controls.

## 1. Introduction

Robust analytical methods to determine food quality attributes, identity and authenticity remain a priority task within their chain supply, to preserve consumers’ protection. Each food matrix possesses intrinsic metabolites related to their manufacturing such as shikimic or caftaric acids in wines [1] or furfural moieties in agave spirits [2,3]. The full set of said primary and/or special metabolites that are intrinsic to the food matrix growth and manufacture at specific conditions and locations can be identified and targeted with the use of omics multisampling technology such as spectroscopic or spectrometric techniques, in order to obtain discriminant observables related to geographical origins, varieties, manufacturing processes, authenticity, amongst others [4,5].

The combination of high-reproducible, non-invasive, rapid and simple-use proton Nuclear Magnetic Resonance Spectroscopy (^1^H-NMR) with Multivariate Statistical Analysis (MSA) for foodstuff metabolomics (foodomics) [6] has emerged over the last decades for the implementation of models to trace food quality, origin, manufacture, or authenticity. ^1^H-NMR metabolomics has been implemented in foodstuff for tracing authenticity and adulterations in acacia honeys by means of sugar profiles [7], discriminations between orange and pulp wash juices as a tool for controlling adulteration practices [8], characterization of the compositional changes in “Tommy Atkins” cultivar mango juices to control postharvest quality [9], quality assessment in traditional balsamic vinegar of Modena according to the ageing process [10] and determination of some compounds defining the originality of Swiss Emmental cheese and discrimination of the studied samples according to their geographical origins [11,12].

Comprising ^1^H-NMR/MSA metabolomics for analysis of the spirits, several reports have emerged over the last decade. Godelman et al. [1] applied the basis of water-to-ethanol multipresaturation during mixing times and recovery delays with 1D-NOESY schemes in approximately 600 German wine samples, for obtaining the data matrix for classifying grape varieties, geographical origins and ageing of five wine-growing areas of southern Germany (Rheinpfalz, Rheinhessen, Mosel, Baden, and Württemberg) with principal component analysis (PCA), linear discrimination analysis (LDA) and multivariate analysis of variance (MANOVA). Noticeable improvements to generate discriminative features within the NMR data matrix of German wine samples were achieved with Independent component Analysis (ICA) combined with LDA [13]. With the use of a T_1_-relaxation filter as a strategy for ethanol suppression instead of water-to-ethanol multi-suppression, proton NMR profiling in combination with PCA, LDA and hierarchical cluster analysis (HCA) was used to discriminate between Italian “Fiano di Avellino” wines produced with the same grape variety, but fermented with commercial or autochthonous yeast starters [14]. Recently, ^1^H-NMR targeted metabolomics were used to discriminate between Chinese wine regions [15] and varieties such as Cabernet Sauvignon, Merlot and Cabernet Gernischt dry red wines [16], as well as different Chardonnay dry white wines treated with different inactive yeasts prior to ageing [17]. Discriminative features came, respectively, from ethyl acetate, lactic acid, alanine, succinic acid, proline, malic acid, gallic acid (red wines) and 2,3-butanediol, ethyl acetate, malic acid, valine, succinic acid, lactic acid, tartaric acid, glycerol, gallic acid, choline, proline, and alanine (white wines) spin systems. Furthermore, specific oenological improvements such as the use of *Hanseniaspora vineae* yeast strains for enhancing aromatic profiles in Spanish Albillo white wines, with respect standard fermentations, was evaluated with both ^1^H-NMR and GC-FID targeted metabolomics [18].

As for wine metabolomics, several improvements have been reported for other commercial alcoholic beverages such as beer [19,20] or whisky [21,22]. In contrast, few reports have emerged comprising the use of NMR/MSA metabolomics for the profiling and/or targeting of North and Central Latin America agave spirits such as tequila or mezcal. Regarding mezcal, its manufacture involves harvesting, cooking, milling, yeast-free fermentation and double distillation of agave’s hearts [23,24,25,26], in agreement with its Nahuatl etymology: *Metlixcalli* which means oven-cooked agaves. Certified mezcals are classified by local regulations by at least three general classes: industrial, artisanal, and ancestral mezcal [27]. Ancestral mezcal is defined as the spirit manufactured by exclusively pit-cooking and mallet or stone-milling the maguey (agave’s common name), with a final distillation comprising a direct heating of the raw fermented material contained in clay pots that are sealed with clay or wooden jackets, by preserving the heating with agave’s bagasse. The use of stainless steel in ancestral mezcal production is not authorized. In contrast, artisanal mezcal can use mechanical shredders for milling, and copper alembics or stainless-steel pots for distillation. Nevertheless, the use of autoclaves for cooking, diffusers to extract juices from cooked maguey and column stills for distillation is prohibited for artisanal mezcal. In Mexico, the appellation d’origine contrôlée (AOC) or controlled designation of origin for mezcals is granted for specific counties in the States of Oaxaca, San Luis Potosí, Puebla, Guerrero, Guanajuato, Durango, Tamaulipas, Zacatecas and Michoacán. Majorly, the commercial mezcals offered from these regions are produced under artisanal practices [28]. All 100% designed mezcals shall not have other sugar source during fermentation rather than those provided by agave sources.

Despite the intrinsic human heritage that implies the fusion between pre-Hispanic fermentations with European distillation processes that derived into ancestral and artisanal mezcals [29], these processes not surprisingly lack standardizations that can guarantee quality and reproducibility amongst batches [24,28]. Mezcals are regulated by the Mexican Official Norm No. NOM-070-SCFI-2016, wherein the essential quality attributes required for compliance are: (i) alcohol by volume [30], (ii) higher alcohol contents [31] (iii) methanol content [32] and ash/dry matter content [33]. Such technical controls are adapted to mezcals’ traditional processes and are not as strict as the compliances required for equivalent spirits such as tequila [34], revealing, in turn, the necessity to find strategies for providing standardization of artisanal or ancestral processes.

The present work uses the combination of ^1^H-NMR spectroscopy and multivariate statistical analysis applied for the first time in a series of artisanal and ancestral mezcals from three different regions and produced from the most representative types of agaves in Mexico. In parallel, it is presented with a scoop, the NMR-MSA profiling of a series comprising two Mexican monovarietal wines, from three different regions and different ageing processes with the same NMR acquisition, pre-processing and multivariate statistical analysis methodologies as with mezcals. Taking, as an advantage, the full set of standardization procedures that exist within the local [35] and international [36] oenological industry that can guarantee its quality amongst batches, the present study proposes the application of the same NMR-MSA workflow in both spirits as a qualitative form to obtain Orthogonal Projections to Latent Structures Discriminant Analysis (OPLS-DA) fingerprints related to geographical origins, raw varieties and specific manufacturing processes amongst individual batches. Finally, standardization procedure differences between mezcals and wines are revealed by the analysis of specific ^1^H NMR-OPLS-DA spectroscopic and statistical features.

## 2. Materials and Methods

### 2.1. Mezcals

A set of 60 types of artisanal, ancestral and industrial mezcal samples from three different regions of Mexico (Oaxaca, Puebla and San Luis Potosí) comprising 3 different types of agaves (*Agave angustifolia Haw*, commonly named as Agave Espadín; *Agave potatorum* Zacc, locally denominated as Agave Tobalá and *Agave salmiana ssp. Crassispina*), all from a 2019 harvest, were used for the present study. Two types of blends were used for the analysis: Blend 1 is a mixture of *Agave potatorum, Agave angustifolia Haw* and *Agave cupreata*; Blend 2 is a mixture of *Agave americana* and *Agave oaxacensis*. Counties per region are labelled as follows: (i) TM: Tlacolula de Matamoros [16°59′10″ N, 96°30′47″ W], (ii) SJT: San Jerónimo Tlacochahuaya [17°00′45″ N, 96°33′08″ W], (iii) TV: Teotitlán del Valle [17°01′13″ N, 96°34′43″ W], (iv) VSV: Villa Sola de Vega [16°29′30″ N, 96°58′34″ W], (v) SFS: San Francisco Sola [16°29′48″ N, 96°57′26″ W], (vi) ZA: Zimatlán de Álvarez [16°52′16″ N, 96°46′42″ W], (vii) SAA: San Agustín Amatengo [16°30′36″ N, 96°47′19″ W], (viii) SBC: San Baltazar Chichicapam [16°45′43″ N, 96°29′23″ W ], (ix) SM: Santiago Matatlán [16°51′60″ N, 96°22′58″ W], (x) CA: Caltepec [18°10′53″ N, 97°28′47″ W], (xi) AT: Atlixco [19°1′25″ N, 98°14′29″ W], (xii) AH: Ahualulco [22°24′1″ N, 101°10′0″ W] and (xiii) MC: Mezquitic de Carmona [22°16′0″ N, 101°6′47″ W]. Full data set comprising processes, regions, counties, and agave species are resumed in Table 1.

### 2.2. Wines

A set of 31 types of Mexican monovarietal wines (Merlot and Cabernet Sauvignon) from 3 different regions (Baja California [32°17′34″ N, 115°5′28″ W], Coahuila [25°27′2″ N, 102°10′37″ W], and Querétaro [20°39′55″ N, 99°53′54″ W]), and different years of vintage (2018 for all Cabernet Sauvignons and both 2017 and 2018 years of vintage for Merlot samples) were used for the present study. Counties per region are labelled as follows: (i) VG: Valle de Guadalupe, (ii) P: Parras and (iii) EM: Ezequiel Montes. Different ageing strategies were used for Merlot and Cabernet Sauvignon samples and coded as [37]: (i) Merlot 2017 aged within a 2017- Tonnellerie d’Aquitaine French barrel, (ii): Merlot 2017 Gran Reserva taken from a 24-months bottled aging, (iii): Merlot 2018 aged within a 2018-Tonnellerie d’Aquitaine French barrel, (iv): Merlot 2018 aged within a 2016-Tonnellerie d’Aquitaine French barrel, (v): Merlot 2018 directly taken from the fermentation tank, (vi): Merlot 2018 aged within a 2016-Boutes French barrel, and (vii): Merlot 2018 aged within a 2018-Demptos American barrel. Full data set comprising type of ageing, regions, counties, and varieties are resumed in Table 2.

### 2.3. ^1^H-NMR Acquisition

Solution-state NMR spectroscopy was carried with a Bruker Avance-III HD spectrometer (Bruker Biospin, Rheinstetten, Germany), operating at 14.1 T of the magnetic field (equivalent to 600 MHz of proton frequency) with a ^1^H/D BBO probehead and a z-gradient. Both mezcal and wine samples were prepared at equivalent conditions for NMR spectroscopy as follows: each mixture containing 540 µL of spirit and 60 µL of deuterium oxide solution 99.9 atom % D that, in turn, contains 0.05 wt% of 3-(trimethylsilyl) propionic-2,2,3,3-d4 acid, sodium salt as an internal reference (CAS No. 7789-20-0) and 0.1% of phosphonate KH_2_PO_4_ (CAS No. 7778-70-0) buffer were prepared and pH adjusted to a value of 3.1 [1,37]. The following NMR schemes were acquired for the full set of wine and mezcal samplings, which, respectively, comprise 31 and 60 samples:(a)Standard direct-excitation one-dimensional proton nuclear magnetic resonance spectra needed to prepare water-to-ethanol off-resonance multipresaturations [37,38,39] were carried out by recording a total of 64 transients, which were collected in 28,844 complex data, with a spectral width of 20 ppm (12,019 Hz), an optimized recovery delay of 6 s and acquisition times of 1.2 s, produced experimental times of 6 min per spectrum. No apodization function was applied during the Fourier Transform.(b)^1^H_water_presat_ NMR: 1D single pulse NOESY experiments with an off-resonance shaped-pulse water presaturation during both relaxation delay (10 s) and mixing times (100 milliseconds) and 8.19 × 10^−4^ W (*vide infra*) and 1.18 × 10^−3^ W power level irradiations, respectively, for wine and mezcal samples, were acquired for all samples at the following conditions: a total of 128 transients were collected within 28K complex data points, with a spectral width of 12,019 Hz and acquisition times of 1.2 s, producing experimental times of 16 min.

### 2.4. ^1^H-NMR Post-Processing and Multivariate Statistical Analysis (MSA)

NMR post-processing for producing the MSA input variables was carried out as follows: ppm calibration and manual phase corrections were conducted with the use of Bruker TopSpin 4.0.8 software. Global and intermediate baseline corrections, least-squares or parametric time warping NMR alignments, variable size bucketing for untargeted profiling and data matrix normalization were carried out with NMRProcFlow software [40]. Scaling and statistical analysis workflow for obtaining the Principal Component (PCA) and the Orthogonal Projections to Latent Structures Discriminant Analysis (OPLS-DA), from the constant sum normalized NMR data matrix, were developed with the BioStatFlow 2.9.2 software. In all cases, T2 Hotelling’s regions depicted by ellipses in score plots of each model define a 95% confidence interval [41]. Supervised OPLS-DA was carried out with Monte-Carlo Cross Validations with 10-test partitions per 100 permutations for testing [42]. In all cases, p-values, R^2^X and Q^2^ statistical parameters that define the quality of each model are expressed [43].

## 3. Results and Discussion

A fair comparison between two different spirits, by means of a ^1^H-NMR/MSA comparable approach, begins with a proper water-to-ethanol multipresaturation proton NMR scheme per case, for obtaining exploitable spectra for multivariate statistical analysis. Appendix A shows a representative set of two standard direct-excitation one-dimensional proton nuclear magnetic resonance spectra of wine and mezcals. Due to their particular alcohol by volume percentage (%ABV) values, the off resonance multipresaturation pulse must be specific to each spirit and applied in the same way between batches. Power levels of 8.19 × 10^−4^ W and 1.18 × 10^−3^ W were found to be optimal respectively for wine and mezcal batches in order to produce ^1^H-NMR spectra with equivalent signal to noise ratios. It is important to highlight that the higher power level for the off resonance multipresaturation pulse in mezcals is intrinsically related to their higher %ABV of around 50% with respect the %ABV of selected wines of around 14.5%.

Figure 1 presents selected spectral widths of the stacked one dimensional {^1^Hwater_presat NMR} spectra of Mexican Cabernet & Merlot wines (A and C, see Table 2) and *Agave angustifolia Haw***,**
*Agave potatorum* Zacc, *Agave salmiana ssp. Crassispina*, *Agave cupreata* mezcals, as well as selected blends (B and D, See Table 1), showing, within the figure, gray buckets, the application of an intelligent binning algorithm amongst aromatic (A and B in Figure 1; 10 ppm ≤ ω^1^H ≤ 5.5 ppm) and aliphatic (C and D in Figure 1; 4.5 ppm ≤ ω^1^H ≤ 0.5 ppm) frequency regions, for the NMR bucketing strategy to produce the reduced data matrices. As observed in Figure 1, all resonances in proximity with to CH_3_ (1.14 ppm) and CH_2_ (3.51 ppm) ethanol signal residuals and to their corresponding ^13^C satellites (±0.1 ppm/±60 Hz at 14.1 Tesla) were not considered for the NMR bucketing strategy. Despite the use of frequency alignment algorithms in both NMR data matrices, such as parametric time warping [44], it can be observed in Figure 1 that some misalignments coexist, mostly in mezcal samples, like the typical acetate singlet at around 2.0 ppm, which could possibly be due to several physicochemical interactions related to the lack of quality controls, particularly during non-standardized ancestral or artisanal agaves’ thermal hydrolysis: the cooking process for liberating fermentable sugars, wherein the final distilled mezcals contain a series of alcohols, aldehydes and organic acids (vide infra) that would challenge the standardization of pH buffering [28]. These imperfections will be reflected in the quality of the produced NMR data matrix submitted to multivariate statistical analysis.

Figure 2, Figure 3 and Figure 4 and Appendix A present the full set of multivariate statistical analyses comprising the Orthogonal Projections to Latent Structures Discriminant Analysis (OPLS-DA) from the constant sum of the normalized NMR data matrix (main text) as well as the prediction accuracy curves as a function of PLS components, permutation tests, Principal Component Analysis (PCA) obtained from the same NMR data matrix and OPLS-DA loading plots of wines’ and mezcal’ data matrices (supporting information material). Unsupervised principal component analysis is generally used for organizing the NMR data matrix and for determining correlations between selected factors (geographical regions, wine’s varieties or agaves’ species and specific ageing or manufacturing process) and outliers (discriminant NMR resonances). In order to maximize separations amongst samples, supervised OPLS-DA was applied to each NMR data matrix. OPLS-DA permits us to obtain optimal information from the dataset by the identification of a more refined multivariate subspace for the maximum group separations by applying Monte-Carlo Cross Validations with a set of partitions per number of permutations (see Section 2.4). At first glance, PCA was applied on both wine and mezcal NMR data matrices, producing plots with equivalent and poor separations between groups described within two-dimensional projections (PC1 = 29%, PC2 around 15 and 17%, Appendix A). For that, OPLS-DA modeling was applied over the full set of wines’ and mezcals’ data matrices for obtaining improved separations amongst factors that allowed pairwise comparisons of discriminative features between wine and mezcals regions, varieties and species and ageing and manufacturing processes. 

Discriminations between wines from Baja California, Coahuila and Querétaro and mezcals from Oaxaca, Puebla and San Luis Potosí by supervised OPLS-DA discriminative analysis are highlighted in Figure 2. Selected oenological regions can be unambiguously discriminated with the use of the selected NMR outlier (R^2^X= 0.999996; Q^2^ = 0.959; *p*-value < 0.004). With less discriminative responses, but still with accurate agreements, agave spirits from Oaxaca, Puebla and San Luis Potosí can be differentiated with the equivalent mezcals’ NMR outlier (R^2^X = 0.8756; Q^2^ = 0.822; *p*-value < 8.48 × 10^−5^). In both cases, the use of 3 PLS components are sufficient to represent observed discriminations of, respectively, 96% and 82%. The higher the amount of discrimination between wines from Baja California, Coahuila and Querétaro relies, not only due to the important set of discriminant loadings obtained from wines’ NMR data matrix, but also from the physical geographical distances between selected regions located, respectively, at the following coordinates: [32°17′34″ N, 115°5′28″ W], [25°27′2″ N, 102°10′37″ W] and [20°39′55″ N, 99°53′54″ W]). In contrast, geographical proximity, mostly between mezcals from Oaxaca (around 16–17° N, 96°22−58′ W) and Puebla (around 18−19° N, 97°–98° W), considerably limits regional discriminations, when compared to wines. Despite the said proximity even with the farthest San Luis Potosí region (around 22°16–24′ N, 101° 6–10′ W), statistically acceptable discriminations are observed with the use of the mezcals’ NMR data matrix.

Discriminations between wines from Merlot and Cabernet Sauvignon wine varieties and mezcals from monovarietal *Agave angustifolia Haw*; *Agave potatorum* Zacc; *Agave salmiana ssp. Crassispina* magueys and selected blends (*Agave potatorum*, *Agave angustifolia Haw* and *Agave cupreata*); (*Agave americana* with *Agave oaxacensis)* by supervised OPLS-DA discriminative analysis are described in Figure 3. Wine varieties can be unambiguously discriminated with the use of the selected NMR outlier (R^2^X = 0.99998; Q^2^ = 0.961; *p*-value < 0.00089). Again, with less discriminative responses, but still with accurate agreements, different agave species can be differentiated with the equivalent mezcals’ NMR outlier (R^2^X = 0.7049; Q^2^ = 0.503; *p*-value < 7.6 × 10^−4^). For wine varieties, the use of 3 PLS components are sufficient to represent observed discriminations above 90%. In contrast, the observed maximum 50% of discrimination between mezcal species is obtained with 2 PLS components, in turn decaying its prediction accuracy when increasing the number of PLS components (Appendix A).

The OPLS-DA supervised model can easily discriminate the San Luis Potosí *Agave salmiana ssp. Crassispina* mezcals from the rest of the species (magenta diamonds in Figure 3). The discriminative analysis also defines two T2 Hotelling’s regions with a 95% confidence interval that slightly delimits *Agave angustifolia* (yellow circles in Figure 3) and *Agave potatorum* (green diamonds Figure 3) species. Furthermore, samples from *Agave potatorum, Agave angustifolia Haw* and *Agave cupreata* mezcal blend rely closely on the *Agave potatorum* (tp, to) subspace, strongly suggesting the contribution of *A. potatorum* to the blend. However, it is not possible to define a subspace of both *Agave americana* and *Agave oaxacensis* blend and *Agave cupreata* monovarietal samples and to distinguish them from *Agave angustifolia* and *Agave potatorum* by diverse possible reasons: (i) the limited loadings that mezcals’ NMR data matrix produces; (ii) the uncertainty that even the mezcal masters claim concerning the authenticity of their magueys due to the lack of an accessible and routine ^13^C-isotopic fractionation analytical method for crassulacean acid metabolism (CAM) plants such as agaves [45,46] throughout the country; (iii) the lack of control comprising the exchange of native agave plants performed by inhabitants of a particular region, mostly from vicinal states such as in Oaxaca and Puebla [47], amongst other reasons.

Discriminations between ageing strategies in wines and the artisanal and ancestral processes in mezcals with supervised OPLS-DA discriminative analysis are presented in Figure 4. Wine NMR outliers that effectively discriminate both oenological regions and varieties produce null discriminations when the OPLS-DA factor relates samples aged in different barrels with respect to samples directly taken from the fermentation tank (R^2^X = 0.409704; Q^2^ = 0.136; *p*-value < 0.177). Interestingly, ancestral and artisanal mezcals can be slightly discriminated (R^2^X = 0.6012; Q^2^ = 0.178; *p*-value < 6.38 × 10^−3^) by means of two 95% confidence T2 Hotelling’s ellipses, wherein ancestral mezcals scores have a range of tp,1 < 0, whilst their artisanal counterparts rely within the tp,1 > 0 range. The use of 3 PLS components are sufficient to represent the observed 18% discriminations.

Finally, OPLS-DA loading plots in Figure 5 resume some key points of the present comparative study as follows: (i) rigorous quality controls applied to Mexican wine samples are responsible of the high ^1^H-NMR spectral reproducibility that in turn allow to the MSA routine, the identification of spectroscopical regions responsible for discriminations; (ii) consequently, an important amount of loadings around the four (to,1/tp,1) quadrants are obtained for wine analysis that allow unequivocal discrimination between selected regions (top left, Figure 5) and varieties (middle left, Figure 5); and (iii) null discriminations amongst different ageing processes in wines is related to a reduced amount of loadings, particularly concentrated in the [-,-] and [+,+] (to,1/tp,1) subspace; iv) to,1 = f(ω^1^H frequency) plots (right, Figure 5) show that discriminant ^1^H-NMR resonances are found all along the wines’ spectral frequency range (red dots). 

Overlay between wine (red dots, Figure 5) and mezcal (green dots, Figure 5) loadings at equivalent (to,1/tp,1/ω^1^H frequency) ranges confirms the validity of comparing mezcals’ regional, species and processes discriminant efficiency as with their wines’ counterparts in the following way: (i) the lack of rigorous quality controls in artisanal and ancestral mezcals is reflected in the ^1^H-NMR data matrix, which in turn produces much less loadings that directly affect the above-mentioned discriminations; (ii) despite said limitations, it is possible to propose regional discriminations regardless their geographical proximity; (iii) difficulty in proposing a species’ of robust discriminant holistic fingerprint relies on the lack of good practices discussed above. Last limitation is expressed within the (to,1/tp,1) subspace, stressed as loadings majorly charged in the tp,1 < 0 quadrants (left middle, Figure 5): (iv) slight discriminations between artisanal and ancestral mezcals are produced due to an important concentration of loadings within the negative (to,1/tp,1) quadrant and v) most of the aliphatic and aromatic/aldehydic ^1^H-NMR discriminant shifts in mezcals correlate with loadings defined at the negative (to,1) quadrants.

Preliminary partial ^1^H NMR assignments for chemical identification of discriminant metabolites in both wine and mezcal spirits presented in Figure 6 and Figure 7 were achieved by comparison to previous reports [1,15,18,28,48] and databases [49]. For both cases, identified discriminant metabolites are shown in independent [1] loading plots as a function of proton chemical shifts. ^1^H NMR allowed the identification of 16 discriminant metabolites in wines (caftaric acid, shikimic acid, fumaric acid, sorbic acid, (β)-glucose, fructose, citric acid, acetoine, malic acid, γ-amino butyric acid (GABA), lactic acid, acetate and some free amino acids such as arginine, isoleucine and valine, see also Table 3) and 11 discriminant metabolites in mezcals (acetaldehyde, 5-substituted furanaldehyde, possibly 5-hydroxymethyl furfural, unsubstituted furfural, 2-furoic acid, (furan-2-yl)-methanol, phenethyl alcohol, phenethyl acetate, ethyl acetate, 1-butanol, 2-butanol and 2-methylpropan-1-ol, see also Table 4).

For wines, one-dimensional ^1^H NMR signal assignment (Table 3) present accurate agreements with respect to previous reports, comprising the typical set of discriminant metabolites such as caftaric acid, sorbic acid, fructose, acetoine, citric acid, malic acid, arginine isoleucine and valine (positive loadings in the present NMR data matrix, see Figure 6), as well as shikimic acid, fumaric acid, glucose, GABA, acetate and lactic acid valine (negative loadings in the present NMR data matrix, see Figure 6). For mezcals, the current report presents the first efforts of a one-dimensional ^1^H NMR profiling based on the chemical shifts (δ, ppm), proton homonuclear scalar couplings (J, Hz) and signal multiplicity assignments of discriminant metabolites’ NMR resonances (Table 4), in comparison to the few available works reporting the major components in mezcals-produced from different agave species with chromatographic analysis [3,26,47]. All identified furane derivatives (2-furfural, 5-substituted furanaldehyde, 2-furoic acid and (furan-2-yl)-methanol), acetaldehyde, aromatic moieties (phenethyl alcohol and phenethyl acetate), ethyl acetate, and n-butanol species, present negative loadings as the fingerprint responsible for OPLS-DA regional, species, and manufacturing processes’ discriminations. In contrast, only 2-methylpropan-1-ol presented a positive loading (see Figure 7).

## 4. Conclusions

The use of a ^1^H-NMR/MSA non-targeted metabolomics workflow is presented for a first-time evaluation of ancestral and artisanal Mexican native mezcals, in order to obtain OPLS-DA regional, species and manufacturing processes holistic fingerprints, wherein their validity and discriminant capacity are supported by Q^2^, R^2^X and p-values statistical parameters that define, in turn, the quality of each model. Limitations of the ^1^H-NMR/MSA mezcals’ discriminant analysis are contrasted with the first-time reported OPLS-DA results obtained for industrialized Mexican wines, whereas the MSA inputs of both systems (NMR acquisition and preprocessing routines) were carefully taken at equivalent conditions. The comparative ^1^H-NMR/MSA produces for both spirits, limited discriminant unsupervised PCA results and trustworthy regional and grape variety or agave species OPLS-DA non-targeted fingerprints, whereas the reduced discriminant capacity of mezcals’ selected regions and agave species with respect to the excellent agreements obtained for their OPLS-DA wine counterparts are attributed to the quality of mezcals’ NMR data matrix, that in turn reflects the lack of rigorous quality controls in mezcals’ production. However, statistically acceptable OPLS-DA results obtained to discriminate between regions and species in mezcals shall be the starting point to encourage producers and local food agencies to use the presented ^1^H-NMR/MSA methodology as a starting point of a national mezcal repository that could serve as an instrument for quality controls within the industry, such as oenological data bases have recently contributed to enhance and control wines’ quality attributes. Null discriminant features encountered for different ageing strategies in wines shall be alleviated by means of modifying the NMR outliers, such as the use of refocusing pulses instead of multipresaturation schemes that could selectively excite the aromatic ^1^H-NMR frequency range for better profiling and targeting the NMR region associated to tannins, differently produced at specific oenological ageing schemes. In the same sense, efforts to retrieve discriminant features for disentangling artisanal from ancestral mezcal production are herein presented, whereas the corresponding OPLS-DA results strongly suggest that by only increasing the dataset will systematically perform the way to distinguish amongst said processes in order to use the proposed ^1^H-NMR/MSA model for quality controls. Once again, the gained knowledge in wine NMR non-targeted metabolomics of the last decade has been applied in one hand to confirm the discriminant metabolites in Mexican wines implied in the OPLS-DA fingerprints related to geographical origins and grape varieties, and on the other hand to apply it in the novel mezcals’ foodomics approach, herein reported. However, further NMR multidimensional methodologies shall be applied in order to confirm the present preliminary ^1^H NMR assignment.

## Figures and Tables

**Figure 1 foods-10-00157-f001:**
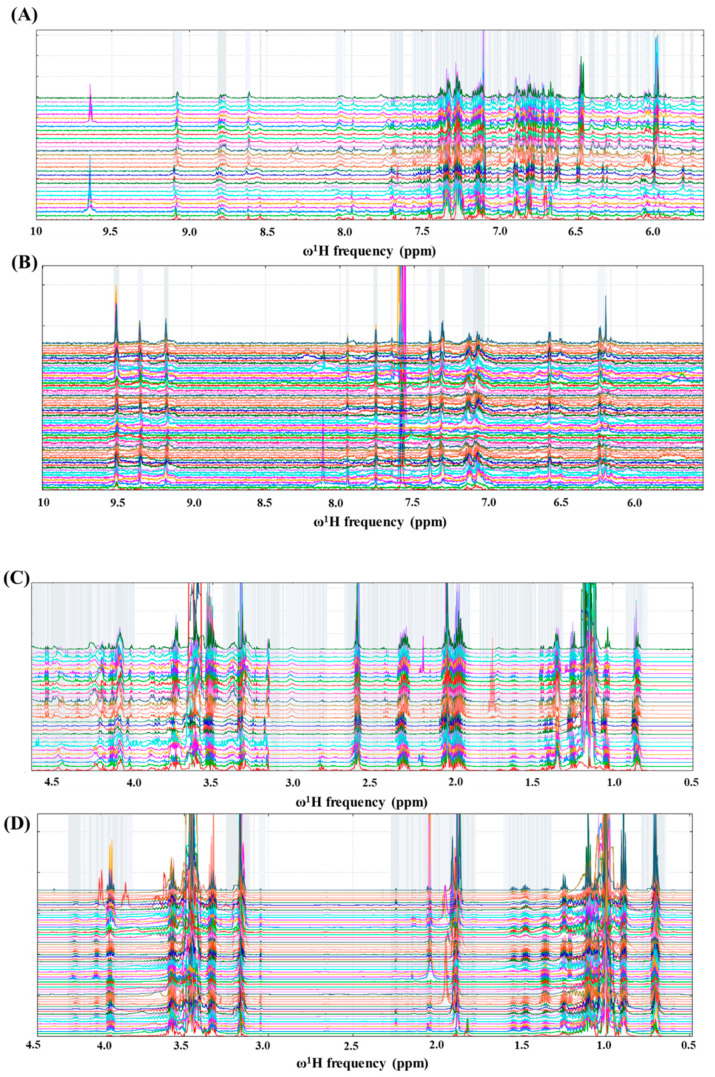
Stacked one dimensional ^1^Hwater_presat NMR spectra of Mexican Cabernet and Merlot wines (**A**,**C**) and *Agave angustifolia Haw*, *Agave potatorum* Zacc, *Agave salmiana ssp. Crassispina*, *Agave cupreata* and selected blends (see Table 1) from Mexican mezcal samples (**B**,**D**). Regions A and B comprise the ^1^H frequency range between 10 and 5.5 ppm, whilst ^1^H NMR expansions in C and D run from 4.5 to 0.5 ppm. Binning strategies for both systems to obtain data dimensionality are highlighted with gray boxes.

**Figure 2 foods-10-00157-f002:**
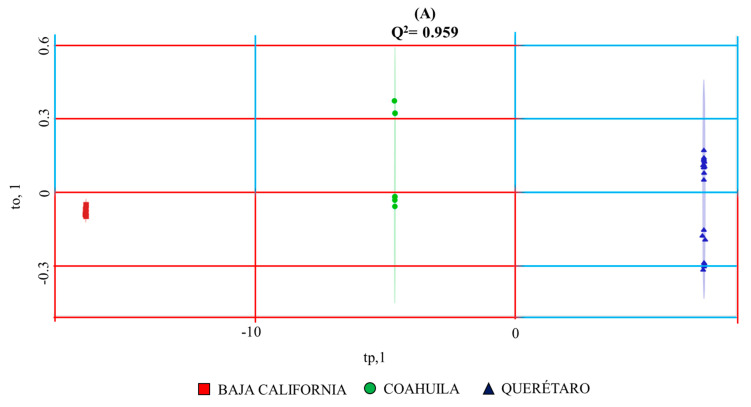
OPLS-DA multivariate statistical analysis score plots generated from ^1^Hwater_presat NMR data matrix (Figure 1) considering the regional factor of wines (**A**) and mezcals (**B**) described in Table 1 and Table 2. T2 Hotelling’s ellipses have a 95% confidence level. Prediction accuracy curves as a function of PLS components, permutation tests, PCA analysis obtained from the same NMR data matrix and OPLS-DA loading plots of wines’ and mezcal’ analysis can be consulted, respectively, in Appendix A.

**Figure 3 foods-10-00157-f003:**
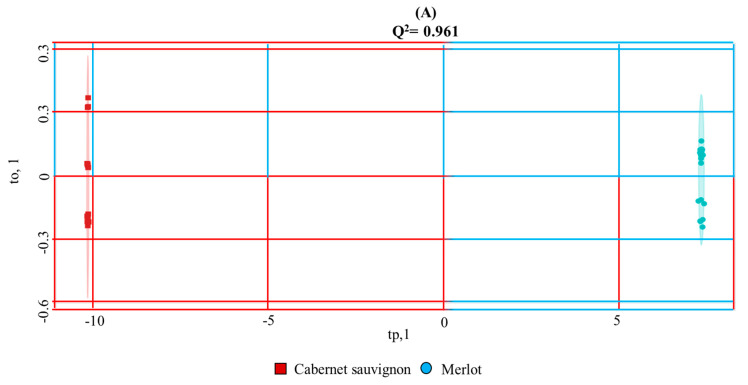
OPLS-DA multivariate statistical analysis score plots generated from ^1^Hwater_presat NMR data matrix (Figure 1) considering wines’ variety (**A**) and agaves’ species (**B**) described in Table 1 and Table 2. T2 Hotelling’s ellipses have a 95% confidence level. Prediction accuracy curves as a function of PLS components, permutation tests, PCA analysis obtained from the same NMR data matrix and OPLS-DA loading plots of wines’ and mezcal’ analysis can be consulted, respectively, in Appendix A.

**Figure 4 foods-10-00157-f004:**
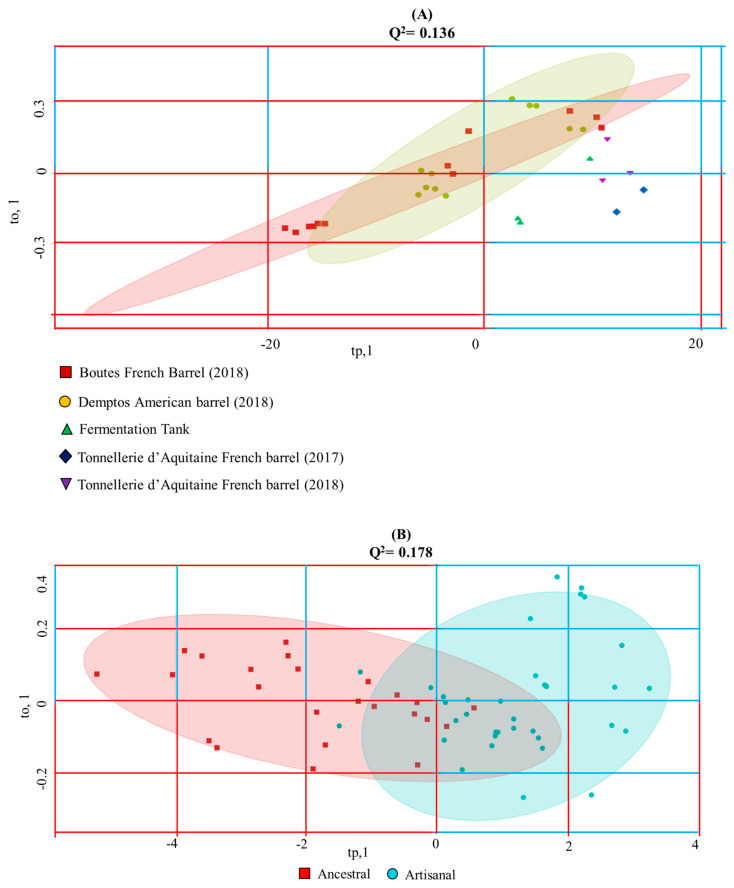
OPLS-DA multivariate statistical analysis score plots generated from ^1^Hwater_presat NMR data matrix (Figure 1) considering wines’ (**A**) and mezcals’ (**B**) ageing and manufacturing processes described in Table 1 and Table 2. T2 Hotelling’s ellipses have a 95% confidence level. Prediction accuracy curves as a function of PLS components, permutation tests, PCA analysis obtained from the same NMR data matrix and OPLS-DA loading plots of wines’ and mezcal’ analysis can be consulted respectively in Appendix A.

**Figure 5 foods-10-00157-f005:**
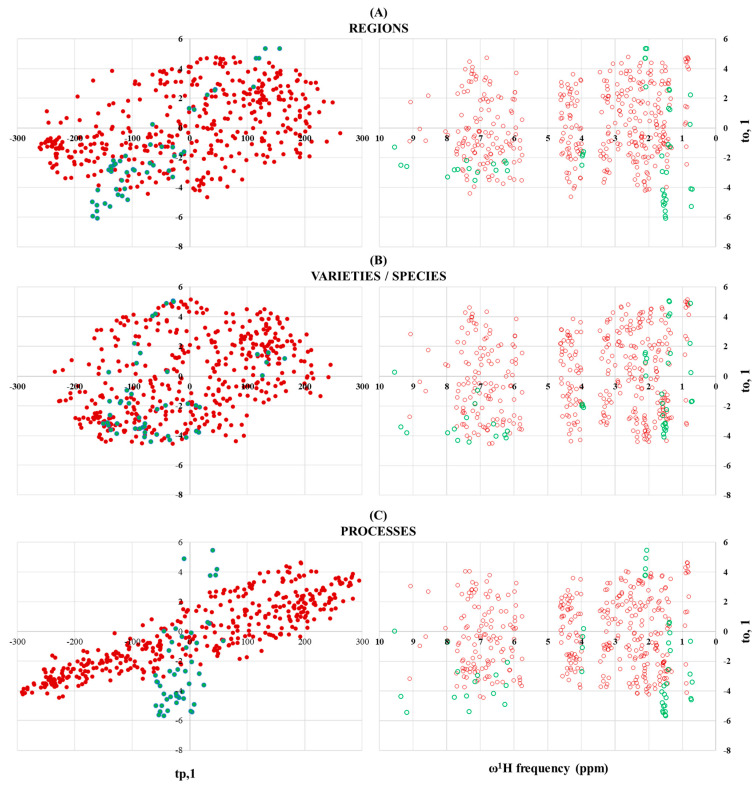
OPLS-DA loading plots of wine (red dots) and mezcal (green dots) regional (**A**), varieties/species (**B**) and processes (**C**) discriminant factors. Orthogonal components (to,1) are expressed as a function of predictive components (tp,1, left plots) and proton chemical shifts (ω^1^H frequency, left plots) from wines’ and mezcals’ NMR data matrices.

**Figure 6 foods-10-00157-f006:**
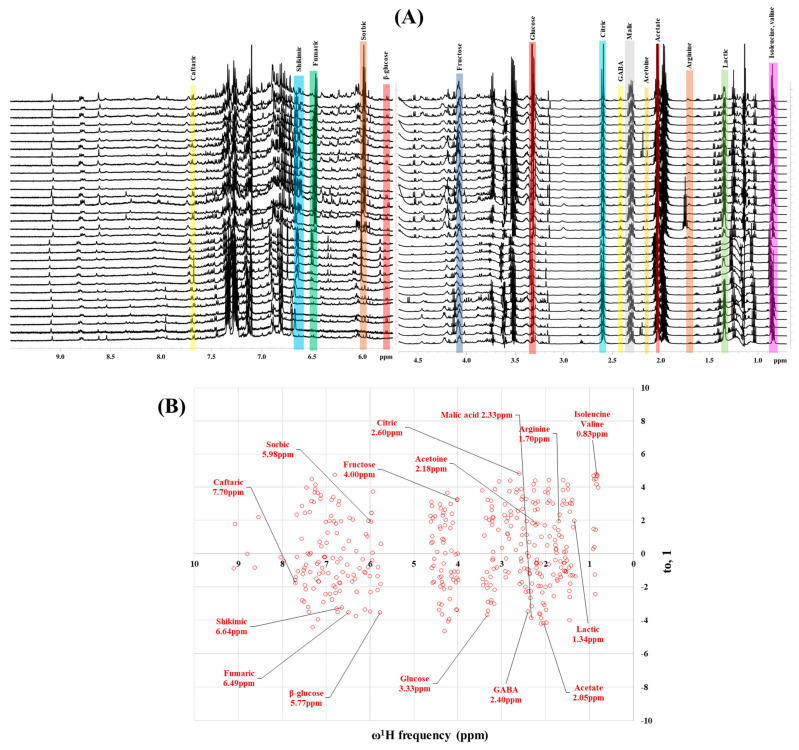
^1^H NMR stacked plot spectra of Mexican Cabernet & Merlot wines (as in Figure 1) with preliminary signal assignments based on literature (**A**) and OPLS-DA [to,1] loading plots as a function of proton chemical shifts with identified metabolites and their assigned chemical shifts (**B**).

**Figure 7 foods-10-00157-f007:**
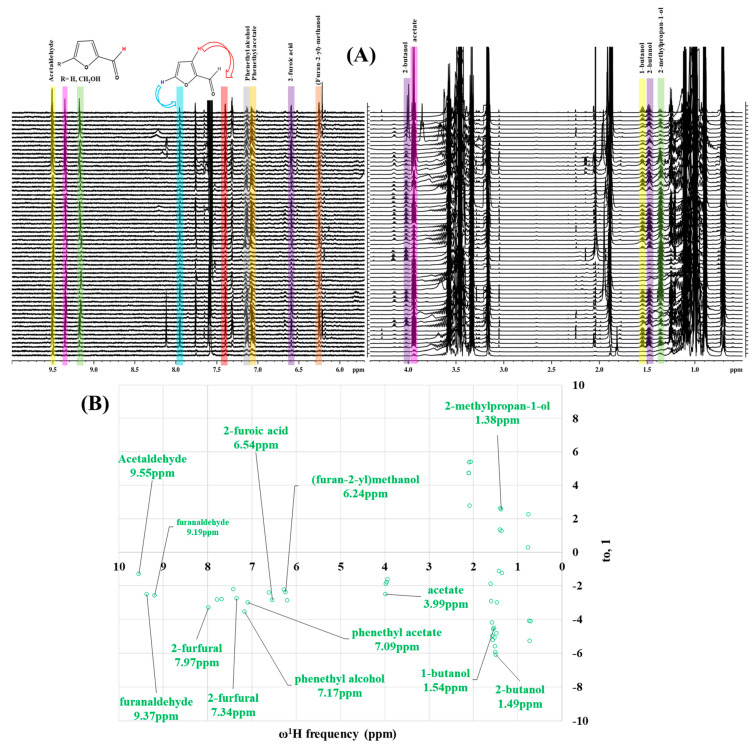
^1^H NMR stacked plot spectra of Mexican mezcals (as in Figure 1) with preliminary signal assignments based on literature (**A**) and OPLS-DA [to,1] loading plots as a function of proton chemical shifts with identified metabolites and their assigned chemical shifts (**B**).

**Table 1 foods-10-00157-t001:** Set of mezcals’ sampling used for NMR/MSA profiling arranged by agave’s species, counties, regions and manufacturing processes.

Mezcal Sampling	Agave Species	County	Region	Process
1	*Agave potatorum*	TM	Oaxaca	Artisanal
2	*Agave angustifolia Haw*	TM	Oaxaca	Artisanal
3	*Agave angustifolia Haw*	SJT	Oaxaca	Artisanal
4	*Agave angustifolia Haw*	TV	Oaxaca	Artisanal
5	*Agave potatorum*	TV	Oaxaca	Artisanal
6	*Agave angustifolia Haw*	VSV	Oaxaca	Ancestral
7	*Agave potatorum*	VSV	Oaxaca	Ancestral
8	*Agave angustifolia Haw*	VSV	Oaxaca	Ancestral
9	*Agave potatorum*	VSV	Oaxaca	Ancestral
10	*Agave potatorum*	VSV	Oaxaca	Ancestral
11	*Agave angustifolia Haw*	VSV	Oaxaca	Ancestral
12	*Agave angustifolia Haw*	VSV	Oaxaca	Ancestral
13	*Agave cupreata*	SFS	Oaxaca	Ancestral
14	Blend No. 1	SFS	Oaxaca	Ancestral
15	*Agave angustifolia Haw*	VSV	Oaxaca	Ancestral
16	*Agave potatorum*	VSV	Oaxaca	Ancestral
17	Blend No. 2	VSV	Oaxaca	Ancestral
18	*Agave angustifolia Haw*	ZA	Oaxaca	Artisanal
19	*Agave angustifolia Haw*	ZA	Oaxaca	Artisanal
20	*Agave potatorum*	SAA	Oaxaca	Artisanal
21	*Agave potatorum*	SBC	Oaxaca	Artisanal
22	*Agave angustifolia Haw*	SM	Oaxaca	Artisanal
23	*Agave potatorum*	CA	Puebla	Artisanal
24	*Agave angustifolia Haw*	AT	Puebla	Artisanal
25	*Agave salmiana ssp. crassispina*	AH	San Luis Potosí	Artisanal
26	*Agave salmiana ssp. crassispina*	MC	San Luis Potosí	Artisanal

**Table 2 foods-10-00157-t002:** Full set of 31 Mexican monovarietal wines’ sampling used for NMR/MSA profiling, arranged by varieties, counties, regions and ageing processes with the year of vintage.

Wine Sampling	Variety	County	Region	Ageing Process (Year of Vintage)
1	Cabernet Sauvignon	P	Coahuila	Demptos American barrel (2018)
2	Cabernet Sauvignon	VG	Baja California	Boutes French barrel (2018)
3	Merlot	EM	Querétaro	T. d’Aquitaine French barrel (2017)
4	Merlot	EM	Querétaro	Demptos American barrel (2018)
5	Merlot	EM	Querétaro	Boutes French barrel (2018)
6	Merlot	EM	Querétaro	T. d’Aquitaine French barrel (2018)
7	Merlot	EM	Querétaro	Fermentation tank (2018)

**Table 3 foods-10-00157-t003:** Assigned ^1^H NMR resonances with chemical shifts (δ, ppm), proton homonuclear scalar couplings (J, Hz) and signal multiplicity of identified metabolites in Mexican wines, showing in each case the assigned shifts within the molecular reduced formula.

**Molecular Reduced Formula (Assigned ^1^H in Red)**	**(δ, ppm), J (Hz), Multiplicity**
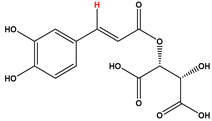 Caftaric acid	δ = 7.70 ppm, J = 15.02 Hz, doublet
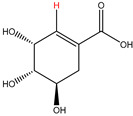 Shikimic acid	δ = 6.64 ppm, J = 8.1 Hz, doublet
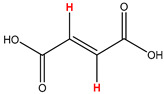 Fumaric acid	δ = 6.49 ppm, J = 15.1 Hz, doublet
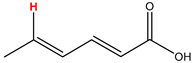 Sorbic acid	δ = 5.98 ppm, multiplet
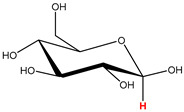 β-glucose	δ = 5.77 ppm, J = 7.55 Hz, doublet
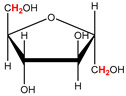 Fructose	δ = 4.00 ppm, multiplet
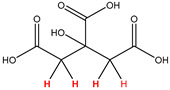 Citric acid	δ = 2.60 ppm, singlet
**Molecular Reduced Formula (Assigned ^1^H in Red)**	**(δ, ppm), J (Hz), Multiplicity**
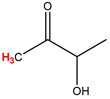 Acetoine	δ = 2.18 ppm, singlet
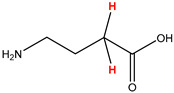 γ-amino butyric acid (GABA)	δ = 2.40 ppm, J= 7.31 Hz, triplet
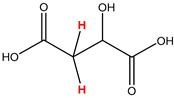 Malic acid (diastereotopic protons)	δ = 2.33 ppm, J = 8.7 Hz, overlaid triplets
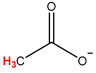 Acetate	δ = 2.05 ppm, singlet
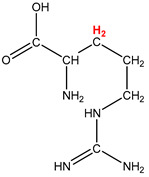 Arginine	δ = 1.70 ppm, J = 6.7 Hz, multiplet
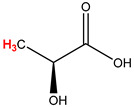 Lactic acid	δ = 1.34 ppm, J = 6.7 Hz, doublet
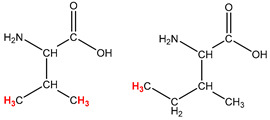 Valine/Isoleucine	δ_max_ = 0.83 ppm, multiplet

**Table 4 foods-10-00157-t004:** Assigned ^1^H NMR resonances with chemical shifts (δ, ppm), proton homonuclear scalar couplings (J, Hz) and signal multiplicity of identified metabolites in Mexican mezcals, showing in each case the assigned shifts within the molecular reduced formula.

**Molecular Reduced Formula (Assigned ^1^H in Red)**	**(δ, ppm), J (Hz), Multiplicity**
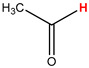 Acetaldehyde	δ = 9.55 ppm, J = 2.82 Hz, quartet
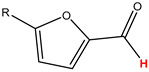 5-substituted furanaldehyde and 2-furfural	δ = 9.37 ppm, singletδ = 9.19 ppm, singlet
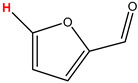 2-furfural	δ = 7.97 ppm, multiplet
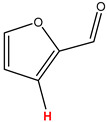 2-furfural	δ = 7.34 ppm, multiplet
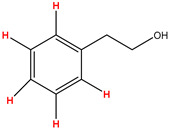 Phenethyl alcohol	δ = 7.17 ppm, J = 7.47 Hz, multiplet
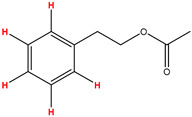 Phenethyl acetate	δ = 7.09 ppm, J = 7.48 Hz, multiplet
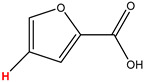 2-furoic acid	δ = 6.54 ppm, J = 3.6 Hz/1.57 Hz, doublet of doublets
**Molecular Reduced Formula (Assigned ^1^H in Red)**	**(δ, ppm), J (Hz), Multiplicity**
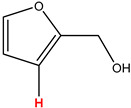 (Furan-2-yl)-methanol	δ = 6.24 ppm, J = 3.3 Hz, doublet
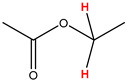 Acetate	δ = 3.99 ppm, J = 7.2 Hz, quartet
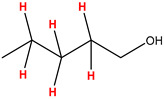 1-butanol	δ = 1.54 ppm, J = 6.67 Hz, multiplet
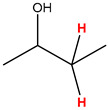 2-butanol	δ = 1.49 ppm, J = 6.67 Hz, multiplet
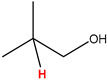 2-methylpropan-1-ol	δ = 1.38 ppm, J = 7.1 Hz, heptuplet

## Data Availability

The data presented in this study are available on request from the corresponding author. The data will be publicly available in national NMR national repositories, currently under progress.

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
