# Peer review of "Comparative NMR Metabolomics Profiling between Mexican Ancestral & Artisanal Mezcals and Industrialized Wines to Discriminate Geographical Origins, Agave Species or Grape Varieties and Manufacturing Processes as a Function of Their Quality Attributes"

_foods, 2021, doi:10.3390/foods10010157_

Round 1

Reviewer 1 Report

Dear Editor, the paper entitled "Comparative 1H NMR metabolomics profiling between Mexican ancestral & artisanal mezcals and industrialized wines to discriminate geographical origins, agave species or grape varieties and specific manufacturing processes as a function of their quality attributes" deals with the NMR acquisition of the proton profiles of 31 wines and 60 mezcals from Mexico. The NMR data were processed and combined with Multivariate Statistical Analysis (MSA) as a tool to discriminate between geographical origins, grape varieties or agave species and manufacturing processes.

The spectroscopic analysis seems accurate and the statistical results are interesting.

A reviewer can hardly raise problems with this type of manuscript. In fact, he cannot recalculate the statistical analysis or check the experimental data. However, the authors should allow researchers, who wish to repeat or improve this method, to verify its correctness. Therefore, I suggest describing in more details the proton spectra of wine and mezcal. For example, showing expansions of the spectral regions and assigning (with certainty or tentatively) the signals to the chemical components.

There are some minor issues:

Wherever it occurs       Please, do not use the conjunction "whereas". I do not easily understand the sentences at lines 76-83, 356-360, 360-363, 363-368, 377-381.

References           the numbering is repeated, the pages are not reported uniformly and the reference 12 is wrong

Line 276               ‘de’ please amend to ‘the’

Author Response

Dear referee,

Many thanks for all your keen observations that have undoubtedly enhanced the quality of the manuscript entitled “Comparative NMR metabolomics profiling between Mexican ancestral & artisanal mezcals and industrialized wines to discriminate geographical origins, agave species or grape varieties and manufacturing processes as a function of their quality attributes

Punctually responding to your observations:

  1. To alleviate your observations related to your suggested discussion about the possible identification of those metabolites that mainly contribute to the sample discrimination, please find that the enclosed reviewed version contains Figures 6 and 7, as well as Tables 3 and 4, containing respectively high-resolution 1H NMR stacked plot spectra of Mexican wines and mezcals with preliminary signal assignments based on literature; OPLS-DA [to,1] loading plots as a function of proton chemical shifts with identified metabolites and their assigned chemical shifts and assigned 1H NMR data with chemical shifts (δ, ppm), proton homonuclear scalar couplings (J, Hz) and signal multiplicity of identified metabolites in Mexican wines and mezcals, showing in each case the assigned shifts within the molecular reduced formula in order to provide robustness and confidence to the preliminary NMR profiling. Please note as well that we specify within the manuscript that this preliminary analysis shall be completed in further studies with the implementation of NMR multidimensional schemes in order to confirm and even increase the list of discriminant metabolites responsible of OPLS-DA regional, species & varieties and manufacturing processes fingerprints.
  2. English changes:
  • Lines 110-111: The cumbersome phrase “…whereas most of commercial mezcals are produced under artisanal practices [26].” Has been changed by “Majorly, the commercial mezcals offered from these regions are produced under artisanal practices [26]”.
  • Lines 118-119: The phrase “whereas the essential quality attributes required for compliance are: i) alcohol by volume [28]…” has been changed by “wherein the essential quality attributes required for compliance are: i) alcohol by volume [28]”
  • Line 353: the phrase: “whereas ancestral mezcals scores have a range of tp,1 < 0 whilst their artisanal counterparts rely within the tp,1 > 0 range.” has been modified to “wherein ancestral mezcals scores have a range of tp,1 < 0, whilst their artisanal counterparts rely within the tp,1 > 0 range.”
  • Line 369-370: The expression “…as follows:” has been modified to “in the following way”
  • Lines 375-376: The phrase “whereas in terms of (to,1 / tp,1) subspace, the last is expressed in loadings majorly charged in the tp,1 < 0 quadrants (left middle, Figure 5);” has been changed to “. Last limitation is expressed within the (to,1 / tp,1) subspace, stressed as loadings majorly charged in the tp,1 < 0 quadrants (left middle, Figure 5)”
  • Lines 378-380: The cumbersome phrase “within the [-,-] (to,1 / tp,1) subspace” Has been improved by: “within the negative (to,1 / tp,1) quadrant”
  • Lines 410-411: The phrase “…whereas their validity and discriminant capacity…” has been changed to “wherein their validity and discriminant capacity”
  • Line 300: The figure caption: “). Regions A and B comprise de 1H frequency range between 10 and 5.5 ppm” has been changed to “). Regions A and B comprise the 1H frequency range between 10 and 5.5 ppm”

  1. References chapter:

 To the best of our knowledge, we have corrected all your provided misleads within the reference chapter, included previous reference 12 (currently reference 14), now completed as: “Mazzei, P.; Spaccini, R.; Francesca, N.; Moschetti, G.; Piccolo, A. Metabolomic by 1H NMR spectrosocopy differentiates “Fiano di Avellino” white wines obtained with different yeast strains. J. Agric. Food Chem. 2013, 61 (45), 10816-10822”.

Again, we are grateful to all your provided observations that have enhanced the quality of our manuscript. We hope that our responses have alleviated your requests

Looking forward to your further decisions, please receive our best regards.

Reviewer 2 Report

In this work, the authors report a complete experimental study about ancestral and artisanal Mexican native mezcals. They use proton NMR spectroscopy combined with multivariate statistical analysis with the aim to discriminate between the different spirits. In this paper, they report the analysis of a non-targeted metabolomics study on such systems for the first time. Although the approach is not innovative, the growing interest towards metabolomic, and in general foodomic approaches involving NMR measurements, gives importance to this kind of study.

The work is indeed of sure interest for people working in the field but some portions are not very clear. The paper need a careful reading and some portions should rewritten. Just as an example of some faults that emerge during reading, see:

  • Strangely, foodomics is not mentioned as the most important and general omic approach for these kinds of analysis. The introduction would benefit of few sentences about that eventually citing some proper literature reviews such as (Food Res. Int. 89, 1085–1094, 2016 and Compr. Rev. Food Sci. Food Saf. 18, 189–220, 2019)
  • Pag. 2 line 72 “was achieved” should be “were achieved”
  • Pag. 3 line 100 “are prohibited” should be “is prohibited”
  • Pag. 4 line 159 “H” should be “1H”
  • Pag. 4 par 2.3.1: the English language should be revised
  • Pag. 4 line 181, same as before
  • Pag 5 lines 199-200 “Due to their intrinsic alcohol by volume percentage (%ABV) differences between them” should be rephrased.
  • Pag. 5 lines 215-216 “Figure 1 that some misalignments coexist mostly in mezcal samples, like the typical acetate singlet at around 2.0 ppm, that could be possibly due to several 217 physicochemical interactions related to the lack of quality controls in mezcals’ production.” please mention which physicochemical interactions could provoke these misalignments.
  • Pag. 5 Lines 235-238 “For that, significantly improved separations for providing pairwise comparisons between wine and mezcals regions, varieties & species and ageing & manufacturing processes, OPLS-DA modeling is applied over all spirits’ data matrix.” Please rephrase
  • The authors should discuss about the possible identification of those metabolites that mainly contribute to the sample discrimination

Author Response

Dear referee,

Many thanks for all your keen observations that have undoubtedly enhanced the quality of the manuscript entitled “Comparative NMR metabolomics profiling between Mexican ancestral & artisanal mezcals and industrialized wines to discriminate geographical origins, agave species or grape varieties and manufacturing processes as a function of their quality attributes

Punctually responding to your observations:

  1. To alleviate your observations related to your suggested discussion about the possible identification of those metabolites that mainly contribute to the sample discrimination, please find that the enclosed reviewed version contains Figures 6 and 7, as well as Tables 3 and 4, containing respectively high-resolution 1H NMR stacked plot spectra of Mexican wines and mezcals with preliminary signal assignments based on literature; OPLS-DA [to,1] loading plots as a function of proton chemical shifts with identified metabolites and their assigned chemical shifts and assigned 1H NMR data with chemical shifts (δ, ppm), proton homonuclear scalar couplings (J, Hz) and signal multiplicity of identified metabolites in Mexican wines and mezcals, showing in each case the assigned shifts within the molecular reduced formula in order to provide robustness and confidence to the preliminary NMR profiling. Please note as well that we specify within the manuscript that this preliminary analysis shall be completed in further studies with the implementation of NMR multidimensional schemes in order to confirm and even increase the list of discriminant metabolites responsible of OPLS-DA regional, species & varieties and manufacturing processes fingerprints.
  2. English changes:
  • Lines 76-78: The phrase “Noticeable improvements to generate discriminative features within the NMR data matrix of German wine samples was achieved with Independent component Analysis (ICA)…” was modified to “Noticeable improvements to generate discriminative features within the NMR data matrix of German wine samples were achieved with Independent component Analysis (ICA)…”
  • Lines 107-108: the phrase “Nevertheless, the use of autoclaves for cooking, diffusers to extract juices from cooked maguey and column stills for distillation are prohibited for artisanal mezcal.” was corrected to “Nevertheless, the use of autoclaves for cooking, diffusers to extract juices from cooked maguey and column stills for distillation is prohibited for artisanal mezcal”
  • English rephrasing of pg. 4 par 2.3.1. and line 181 has been done (modifications are highlighted in red color)
  • English rephrasing of pg. 5 lines 199-200 “due to their intrinsic alcohol by volume percentage (%ABV) differences between them” has been modified to “Due to their particular alcohol by volume percentage (%ABV) values, the off resonance multipresaturation pulse must be specific per spirit and applied in the same way between batches”
  • For attending your request to mention physicochemical interactions that provoke NMR signal misalignment, we have completed the text as follows: “some misalignments coexist mostly in mezcal samples, like the typical acetate singlet at around 2.0 ppm, that could be possibly due to several physicochemical interactions related to the lack of quality controls particularly during non-standardized ancestral or artisanal agaves’ thermal hydrolysis: the cooking process for liberating fermentable sugars, wherein the final distilled mezcals contains series of alcohols, aldehydes and organic acids (vide infra) that would challenge the standardization of pH buffering [28].”
  • The cumbersome phrase “For that, significantly improved separations for providing pairwise comparisons between wine and mezcals regions, varieties & species and ageing & manufacturing processes, OPLS-DA modeling is applied over all spirits’ data matrix.” has been restructured to: “For that, OPLS-DA modeling was applied over the full set of wines’ and mezcals’ data matrices for obtaining improved separations amongst factors that allowed pairwise comparisons of discriminative features between wine and mezcals regions, varieties & species and ageing & manufacturing processes”

  1. References chapter:

Many thanks for remarking the importance of defining foodomics as a central concept that outlines the present manuscript, as well as the excellent references to reinforce said concept. Please note that in the reviewed version of the manuscript, said references has been included as:

6. Hatzakis, E. Nuclear magnetic resonance (NMR) spectroscopy in food science: A comprehensive review. Compr. Rev. Food Sci. Food Saf. 2019, 18, 189–220

12. Corsaro, C.; Cicero, N.; Mallamace, D.; Vasi, S.; Naccari, C.; Salvo, A.; Giofrè, S. V.; Dugo, G. HR-MAS and NMR towards foodomics, Food Res. Int. 2016, 89, 1085-1094

Whereas we have as well used the concept of foodomics within the abstract (line 25), Introduction (lines 57-60) and Conclusions (lines 464-468).

Again, we are grateful to all your provided observations that have enhanced the quality of our manuscript. We hope that our responses have alleviated your requests

Looking forward to your further decisions, please receive our best regards.

Reviewer 3 Report

In the Introduction, article includes references, from number 11 to 16, to authors that use Nuclear Magnetic Resonance Spectroscopy (1H-NMR) with assignment on which the difference of individual samples is based with respect to their composition. In this article the assignment of 1H-NMR spectra was not implemented. The entire statistical analysis was conducted only on the shift of H-NMR without the assigmment of the analyte. It is not clear why there is a comparison of Mezcals and wines in the same article as these are two different categories of agricultural products.

Authors must include information about assignation of compounds and conduct statistical analysis with respect of different chemical compounds.

Author Response

Dear referee,

Many thanks for all your keen observations that have undoubtedly enhanced the quality of the manuscript entitled “Comparative NMR metabolomics profiling between Mexican ancestral & artisanal mezcals and industrialized wines to discriminate geographical origins, agave species or grape varieties and manufacturing processes as a function of their quality attributes

Punctually responding to your observations:

  1. To alleviate your request for developing signal assignments for the identification of those metabolites that mainly contribute to the sample discrimination, as well as the respective statistical analysis, please find that the enclosed reviewed version contains Figures 6 and 7, as well as Tables 3 and 4, containing respectively high-resolution 1H NMR stacked plot spectra of Mexican wines and mezcals with preliminary signal assignments based on literature; OPLS-DA [to,1] loading plots as a function of proton chemical shifts with identified metabolites and their assigned chemical shifts and assigned 1H NMR data with chemical shifts (δ, ppm), proton homonuclear scalar couplings (J, Hz) and signal multiplicity of identified metabolites in Mexican wines and mezcals, showing in each case the assigned shifts within the molecular reduced formula in order to provide robustness and confidence to the preliminary NMR profiling. Please note as well that we specify within the manuscript that this preliminary analysis shall be completed in further studies with the implementation of NMR multidimensional schemes in order to confirm and even increase the list of discriminant metabolites responsible of OPLS-DA regional, species & varieties and manufacturing processes fingerprints.
  2. In an attempt to clarify the reasons for comparing mezcals and wines in the present manuscript, we share with you the following main concepts that are also stressed within the entire manuscript in its reviewed version:

To the best of our knowledge, the present comparative study shows for the first time a NMR/OPLS-DA supervised discriminant non-targeted metabolomics analysis of ancestral and artisanal mezcals, from three of the most representative regions of Mexico that possess in turn their local appellations d'origine contrôlée certificates. Also, novel NMR/OPLS discriminant analysis of Mexican industrialized wines are as well presented. Both NMR acquisition and pre-processing schemes were carefully taken at equivalent conditions in order to have fair comparisons amongst both type of spirits. In both cases, similar factors for multivariate statistical analysis were consider for obtaining discriminant holistic fingerprints: Regions, Grape varieties or Agave species and specific ageing or manufacturing processes.

The main objective of the present comparative study is to demonstrate by NMR/MSA non-targeted metabolomics, in one hand its confirmed robustness to discriminate between oenological regions and wines from a specific monovarietal strain (such as extensively reported since almost a decade ago for wines coming from other regions such as Western Europe or more recently in China, see References), by specifically enhancing their industrial manufacturing origin. In the other hand, the present work stresses the limitations to apply the equivalent NMR/MSA workflow over a set of samples that were manufactured over a non-professionalized scheme, such as ancestral and artisanal mezcals. Once peer reviewed and validated by referees, we are looking forward to diffuse our comparative study within local producers and food officers as a starting point towards the professionalization of the mezcal industry, by demonstrating the possibilities to obtain robust regional, species and manufacturing processes discriminations between mezcals, such as the robust results herein presented for Mexican industrialized wines, in agreement to current recommendations that Intergovernmental Organizations and Control Agencies are promoting for the use of  NMR/MSA technology as quality compliances for tracking : standard and degradation parameters, fermentation products, polyphenols, amino acids, geographical origins, appellations d'origine contrôlée and type of monovarietal strains in wines.

Again, we are grateful to all your provided observations that have enhanced the quality of our manuscript. We hope that our responses have alleviated your requests

Looking forward to your further decisions, please receive our best regards.

Round 2

Reviewer 3 Report

After the revision made by the authors and their responses to my previous comments, I recommend acceptance of this article.